# Preterm Infants Harbour a Rapidly Changing Mycobiota That Includes *Candida* Pathobionts

**DOI:** 10.3390/jof6040273

**Published:** 2020-11-09

**Authors:** Stephen A. James, Sarah Phillips, Andrea Telatin, David Baker, Rebecca Ansorge, Paul Clarke, Lindsay J. Hall, Simon R. Carding

**Affiliations:** 1Gut Microbes and Health, Quadram Institute Bioscience, Norwich Research Park, Norwich NR4 7UQ, UK; steve.james@quadram.ac.uk (S.A.J.); sarah.phillips@quadram.ac.uk (S.P.); andrea.telatin@quadram.ac.uk (A.T.); david.baker@quadram.ac.uk (D.B.); rebecca.ansorge@quadram.ac.uk (R.A.); 2Neonatal Intensive Care Unit, Norfolk and Norwich University Hospitals NHS Foundation Trust, Norwich NR4 7UY, UK; paul.clarke@nnuh.nhs.uk; 3Norwich Medical School, University of East Anglia, Norwich NR4 7TJ, UK; 4Ziel—Institute for Food and Health, Technical University of Munich, 85354 Freising, Germany

**Keywords:** mycobiome, GI tract, preterm infant, early life, pathobiont, fungi, yeast, *Candida parapsilosis*

## Abstract

Fungi and the mycobiome are a fundamental part of the human microbiome that contributes to human health and development. Despite this, relatively little is known about the mycobiome of the preterm infant gut. Here, we have characterised faecal fungal communities present in 11 premature infants born with differing degrees of prematurity and mapped how the mycobiome develops during early infancy. Using an ITS1 sequencing-based approach, the preterm infant gut mycobiome was found to be often dominated by a single species, typically a yeast. *Candida* was the most abundant genus, with the pathobionts *C.*
*albicans* and *C.*
*parapsilosis* highly prevalent and persistent in these infants. Gestational maturity at birth affected the distribution and abundance of these *Candida*, with hospital-associated *C.*
*parapsilosis* more prevalent and abundant in infants born at less than 31 weeks. Fungal diversity was lowest at 6 months, but increased with age and change of diet, with food-associated *Saccharomyces*
*cerevisiae* most abundant in infants post weaning. This study provides a first insight into the fungal communities present within the preterm infant gut, identifying distinctive features including the prominence of pathobiont species, and the influence age and environmental factors play in shaping the development of the mycobiome.

## 1. Introduction

The human gastrointestinal (GI) tract harbours a complex microbial ecosystem comprising a vast number and array of bacteria, archaea, fungi, protozoa and viruses that constitute the intestinal microbiota [1,2]. Most of what we currently know regarding the contribution the enteric microbiota makes to human health and disease comes from the analysis of the prokaryome. Various studies of bacterial constituents of the microbiota have provided evidence for their ability to influence various physiological processes including digestion, micronutrient biosynthesis, energy homeostasis, pathogen colonisation resistance, and maintenance of epithelial barrier integrity [3], and the development and functional maturation of the host immune system [4]. Disruption of the prokaryome (dysbiosis) in early life can have a profound and long-lasting impact on health later in life, including the development of non-communicable illnesses, such as atopy and allergic disease (e.g., asthma), and metabolic disorders [5,6,7,8].

By comparison, far less is known about the mycobiome and fungal residents of the GI tract, and their involvement in human health and disease. This is in part related to their low abundance [9]. In a recent metagenomic study, fungi were estimated to make up approximately 0.1% of the microbiota in the healthy adult GI tract [1]. In considering that fungi are larger in volume than bacteria, they are likely to constitute a sizeable proportion of the intestinal microbiota biomass [10]. Thus, despite their low abundance, fungi can have significant impacts on human health [11]. The yeast *Saccharomyces boulardii*, a probiotic strain of *S. cerevisiae*, is frequently prescribed as an effective treatment for antibiotic-associated diarrhoea and irritable bowel syndrome (IBS) [12,13,14,15]. In contrast, fungal dysbiosis and reduced fungal diversity, coupled with increased levels of *Candida parapsilosis* and *Pichia jadinii* (*C. utilis*) has been observed in paediatric cases of inflammatory bowel disease (IBD) [16]. IBD pathogenesis has also been correlated with an increased Basidiomycota/Ascomycota ratio and increased abundance of *C. albicans* [17]. Fungal dysbiosis has also been reported in colorectal adenoma, and in metabolic syndrome and obesity [17,18,19]. Fungal pathobiont overgrowth leading to infection, often as a result of prolonged antibiotic exposure, also poses a significant health risk, especially in immune deficient individuals, such as those undergoing chemotherapy and solid organ transplantation [11].

Preterm infants represent a highly vulnerable and ‘at risk’ patient group, having endured disrupted fetal development, with the final weeks/months of development occurring ex utero in a neonatal intensive care unit (NICU). A shortened gestational period, shortened labour and possible delivery by caesarean section are all factors that can contribute to reduced exposure to maternal vaginal and enteric microbiotas [20,21]. Indeed, the bacterial composition of the preterm gut is distinct, characterised by a reduced diversity and dominance of facultative anaerobes (e.g., *Enterococcus* and *Staphylococcus* spp.) [22,23]. Preterm infants also have an underdeveloped immune system and an intestinal epithelial barrier with increased permeability. Consequently, this makes them highly susceptible to nosocomial infection, especially from the ‘ESKAPE’ pathogens, a group of virulent and multi-drug resistant bacteria which include *Enterococcus faecium*, *Klebsiella pneumoniae* and *Staphylococcus aureus* [24]. In addition, these infants are at increased risk of developing late-onset sepsis and necrotising enterocolitis [25,26,27]. Broad-spectrum antibiotics are also frequently administered to preterm infants in their first weeks of life. However, an adverse effect of such treatment is increased susceptibility to fungal infections resulting in significant morbidity and mortality [28]. These can arise either from overgrowth of resident fungal pathobionts (e.g., *C. albicans*) or from horizontally acquired species within the hospital environment (e.g., *C. parapsilosis*) [29,30,31,32,33,34]. To reduce the risk of such infections, prophylactic antifungals are routinely administered in the first weeks of life to the highest risk preterm babies, specifically those born at less than 27 weeks’ gestation and/or weighing less than 750 g, while they have an indwelling central venous catheter.

In the present study, we developed laboratory protocols and a fungal analysis bioinformatics pipeline to characterise the faecal mycobiome of a small cohort of young infants all born with differing degrees of prematurity, over the course of a 12-month period in early infancy. The goal was to gain a first insight into the composition and diversity of the preterm infant gut mycobiome, and how it changes between the first 6 to 18 months’ postnatal life; an important developmental window in early infant life.

## 2. Materials and Methods

### 2.1. Study Participants and Samples

The infant cohort comprised 11 infants, 5 females and 6 males, aged less than 2-years old. All were born preterm at the Norfolk and Norwich University Hospital (NNUH), with gestational ages ranging from 25 weeks (extremely preterm) to 36 weeks (late preterm), with a median age of 30 + 4 weeks. All 11 infants were participants in the BAMBI study (Reference no: ‘2012/2013—42HT’), a longitudinal microbiota profiling study to determine the impact of *Bifidobacterium* and *Lactobacillus* supplementation on the bacterial composition of the preterm infant gut [35]. All subjects gave their informed consent for inclusion before they participated in the study. The study was conducted in accordance with the Declaration of Helsinki, and the protocol was approved by the Quadram Institute Bioscience Ethics Committee and in accordance with protocols by the National Research Ethics Service (NRES) approved UEA/QIB Biorepository (License no: 11208). Clinical metadata for the infants including gender, gestational age at birth, birthweight, mode of delivery, and postnatal antimicrobial treatment, along with whether they received breastmilk and/or formula during their stay at the NNUH are provided in Appendix A.

### 2.2. DNA Extraction

Infant faecal samples were collected at three time points; 6 months, 12 months and 18 months and were aliquoted and stored at −70 °C prior to processing and DNA extraction. Total microbial DNA was extracted from ~200 mg of faeces using the FastDNA Spin Kit for Soil (MP Biomedicals, Irvine, CA, USA) and following the manufacturer’s protocol. In addition, all samples were homogenized using a FastPrep-24 benchtop instrument (MP Biomedicals) at 6.0 m/s for 3 × 1 min with 5 min resting intervals on ice. Extracted DNA was quantified and quality checked using the Qubit 3.0 fluorometer and associated Qubit dsDNA BR Assay Kit (Thermo Fisher, Waltham, MA, USA), as well as DNA visualisation by stained agarose gel electrophoresis (1% agarose gel stained with Midori Green Direct DNA Stain). A flowchart outlining the DNA extraction protocol is shown in Appendix A (left hand panel).

### 2.3. ITS1 Amplification, Library Preparation and Sequencing

The fungal ITS1 region was amplified from 100 ng of faecal DNA by PCR using the ITS1F and ITS2 primer set [36,37], with each primer modified at the 5′ end to include an Illumina adapter tail using the following amplification conditions: 94 °C for 5 min; 35 cycles of 92 °C for 30 s, 55 °C for 30 s, and 72 °C for 45 s; and a final extension of 72 °C for 5 min. Amplification reactions were set up in duplicate for each faecal DNA sample, and positive and negative controls were also included in each PCR run (see Section 2.5). Following ITS1 PCR, a 0.7× SPRI purification using KAPA Pure Beads (Roche, Wilmington, MA, USA) was performed and the purified DNA was eluted in 20 µL of EB buffer (10 mM Tris-HCl). In a second PCR, library index primers were added using a Nextera XT Index Kit v2 (Illumina, Cambridge, UK) and following the following amplification conditions: 95 °C for 5 min: 10 cycles of 95 °C for 30 s, 55 °C for 30 s, and 72 °C for 30 s; and a final extension of 72 °C for 5 min. Following PCR, libraries were quantified using the Invitrogen^™^ Quant-iT dsDNA high sensitivity assay kit (Thermo Fisher) and run on a FLUOstar Optima plate reader (BMG Labtech, Aylesbury, UK). Libraries were pooled following quantification in equal quantities. The final pool was SPRI cleaned using 0.7× KAPA Pure Beads, quantified on a Qubit 3.0 fluorometer and run on a High Sensitivity D1000 ScreenTape (Agilent Inc., Santa Clara, CA, USA) using the Agilent Tapestation 4200 to calculate the final library pool molarity. The pool was then run, at a final concentration of 8 pM, on an Illumina MiSeq instrument using the MiSeq^®^ Nano v2 (2 × 250 bp) Kit (Illumina). All sequencing was performed at the Quadram Institute Bioscience, Norwich. The raw data were analysed locally on the MiSeq instrument using MiSeq reporter.

### 2.4. Mycobiome Characterisation

Illumina MiSeq reads were analysed using a hybrid pipeline using USEARCH [38] and QIIME 2 [39]. After removal of locus-specific primers and reads with ambiguous bases using fastp 0.20.0 [40], the remaining reads were pooled, quality filtered and dereplicated using USEARCH. Identification of representative sequences was performed using the UNOISE3 algorithm [41], to produce a set of amplicon sequence variants (ASVs). A feature table was produced with USEARCH. Representative sequences and the feature table were imported as QIIME 2 artifacts. QIIME 2 was used to assign the taxonomy using the UNITE Fungal ITS database (release 04.02.2020) [42], to align ASVs using mafft [43] via “q2-alignment”, and to construct a phylogeny with fasttree2 [44], using “q2-phylogeny”. Community composition was assessed using R 3.6.2 [45] and the R package phyloseq [46]. The feature table, taxonomic classification, phylogeny and metadata were imported into R to create a phyloseq object. Every ASV with a zero count in all samples was removed to assess alpha diversity measures. Community composition and beta-diversity measures were assessed after removing samples with <7500 counts, transforming counts to relative abundances and removing taxa with <5% relative abundances in all samples. A flowchart outlining the main steps in the mycobiome analysis pipeline is shown in Appendix A (right hand panel)

The raw Illumina ITS1 sequence data produced by the present study have been deposited at the NCBI Short Reads Archive (SRA), under BioProject accession number PRJNA627184. All scripts used in this study are available at https://github.com/quadram-institute-bioscience/bambi-its/.

### 2.5. Inclusion of Controls

Controls were included at each stage of this study. During DNA extraction, an empty bead-beating tube was included and treated exactly the same as for the tubes containing faecal samples, and was quantified and visualised similarly. This extraction control was included in the initial amplicon PCR to assess that no ITS1 amplicon was produced. Negative (PCR dH_2_O) and positive controls (0.01 ng *C. albicans* DNA) were included for all PCR reactions. Libraries were also prepared from the DNA extraction control and from *C. albicans* DNA and were used as pipeline controls in the downstream bioinformatic analyses.

## 3. Results

### 3.1. ITS1 Based Mycobiome Profiling and the Infant Gut Mycobiome

We developed a culture-independent protocol and bioinformatics pipeline to amplify and sequence the fungal internal transcribed sequence (ITS) 1 region located between the 18 S and 5.8 S rRNA genes to gain a better insight into the composition and diversity of the infant gut mycobiome in early life, and how it changes over time. The protocol and pipeline was used to characterise the faecal fungal communities present in 11 healthy young infants comprising five females (F1 to F5) and six males (M1 to M6), all of whom were born premature (see Appendix A). Faecal samples were collected over a 12-month period, at 6, 12 and 18 months. Following collection, samples were aliquoted and stored at −70 °C until required. Although culture-dependent protocols were also employed in this study, prolonged sample freezing was found to prevent the isolation of viable fungi. A total of 29 faecal samples were obtained (for four infants only two samples were available) for ITS1 sequencing. A total of 1,609,183 quality-trimmed ITS1 reads were obtained, ranging from 8924 for infant F3, 18-month sample (F3_18) to 130,247 for M2_12, with a sample average of 55,489 reads (see Datasheet S1). One hundred and twelve fungal taxa were detected, with the majority (86%) classified to the genus level or below. Two taxa could not be assigned at the phylum level or below, and were classified as ‘Unidentified’ (see Datasheet S1). The number of fungal taxa detected in each infant sample ranged from five to 37, with a sample average of 15.8 (Appendix A). A significant proportion of the fungi detected were infant and sample time point-specific (47/112; 42.0%).

At the phylum level, >98% of fungi belonged to either Ascomycota or Basidiomycota. Sixty taxa were ascomycetes, 44 of which could be identified to species level, while 50 taxa were basidiomycetes, and 34 were identified to species level (Appendix A). Ascomycota was the predominant phylum, accounting for 60% of all ITS1 reads, and was dominant in many of the samples (Figure 1a). Ascomycetes were the most abundant fungi at all collection timepoints (Appendix A).

*Candida* was present in all 29 infant samples (Appendix A). In some samples, such as the 6-month samples from infants F1, M1, M3 and M5, *Candida* was the predominant genus (Figure 1b). Other prevalent genera, of varying abundance, included *Davidiella* (23), *Penicillium* (21), *Aspergillus* (18), *Saccharomyces* (15) and *Debaryomyces* (14) (Appendix A), some of which dominated individual samples (e.g., *Debaryomyces*, F1_12; *Saccharomyces*, F3_18) (Figure 1b). Amongst the Basidiomycota, *Baeospora* (e.g., *B. myosura*) was the most abundant and prevalent genus, present in all but two samples (Figure 1b; Appendix A). Five other prevalent basidiomycetous genera were *Malassezia* (13), *Rhodotorula* (12), *Filobasidium* (9), *Naganishia* (8) and *Vishniacozyma* (8). However, all of these genera were present in low abundance, with individual mean abundances of 0.2 to 2.9% (*Filobasidium* and *Naganishia*, respectively) with many basidiomycetous genera occurring at abundances of <0.2% (Appendix A).

Among the ten most abundant genera of either phylum, six were yeast genera (i.e., *Candida*, *Debaryomyces*, *Meyerozyma*, *Naganishia*, *Rhodotorula* and *Saccharomyces*) (Figure 1b). Of the 78 fungi resolved to species level, six of the most abundant were yeasts. Moreover, these yeasts were subdivided into two distinct groups, human-associated (e.g., *Candida albicans*) and food-related (e.g., *Saccharomyces cerevisiae*) (Appendix A).

### 3.2. Prevalence of Opportunistic Candida Pathogens

*Candida* accounted for 28.8% of all fungal reads, and was present in every infant sample, ranging from 0.02 to 97.1% abundance (in F4_12 and M1_06, respectively). In total, six *Candida* species were identified, including most notably *C. albicans*, *C. metapsilosis*, *C. parapsilosis* and *C. tropicalis*, all of which are human-associated and fungal pathobionts [47]. In the case of the other two species, namely *C. natalensis* and *C. railenensis*, both were discounted from further analyses based on their inability to survive above 30 °C [35].

Of the four human-associated *Candida*, at least one species was present in every infant sample with 79% of the samples containing two or more species with *C. albicans* and *C. parapsilosis* found in 17/23 (74%) samples (Figure 2). Overall, *C. parapsilosis* was the most abundant (mean of 13.3%) and prevalent (90%). Only three infants did not carry *C. parapsilosis* at all timepoints (Figure 2). *C. albicans* was the second most abundant (mean of 10.9%) and prevalent species (66%). In contrast, *C. metapsilosis* and *C. tropicalis* were less common, present in four and eight samples, respectively. Despite its low prevalence, *C. metapsilosis* was nevertheless the most abundant species (70.1%) in the 6-month sample of infant M5 (Figure 2). This infant was unique as he was the only member of the cohort categorized as ‘high risk’, based on his gestational age at birth (<27 weeks) and a birthweight of <750 g. Such ‘high risk’ infants routinely receive prophylactic antifungals while fitted with an indwelling central venous catheter to help reduce the risk of fungal infection, and fluconazole was the prophylactic antifungal administered to this infant (from day 7 to 36; see Appendix A).

All infants were born premature and their gestational ages (GA) at birth ranged from 25 to 36 weeks (Appendix A). When the longitudinal samples were re-ordered from earliest to latest GA, there were clear differences between the distribution and abundance of the four *Candida* species; most notably between *C. albicans* and *C. parapsilosis* (Figure 3). For the six infants born at <31 weeks, *C. parapsilosis* was the more prevalent, present in 14/16 samples (~87%), compared to 9 samples (~56%) for *C. albicans*. Furthermore, *C. parapsilosis* was also more abundant than *C. albicans* in these infants (12.6% versus 2.7%, respectively). In contrast, for the five infants born at 31 weeks or later, while both species were present in a comparable number of samples (*C. albicans*, 10; *C. parapsilosis*, 12), *C. albicans* was more abundant (*C. albicans*, 21.0%; *C. parapsilosis*, 14.1%). Whereas the mean relative abundance of *C. parapsilosis* remained at a similar level between the two infant groups (GA: <31 weeks, 12.6%; ≥31 weeks, 14.1%), it increased ~8-fold for *C. albicans* (GA: <31 weeks, 2.7%; ≥31 weeks, 21.0%).

### 3.3. Fungal Species Persistence

Fungal persistence in serial samples is indicative of successful colonisation. To investigate this further, longitudinal samples from each individual infant were examined focusing on taxa identified to species level. A total of 23 species were identified of which nine appeared infant-specific (e.g., *Aspergillus ruber*, infant M1), with 14 re-occurring in two or more infants (e.g., *Debaryomyces hansenii*; infants F1, F3, M1, M3 and M6) (see Appendix A).

By applying the criteria of survivability and growth at body temperature, 14 fungi were discounted as transients based on their inability to grow at 37 °C. The remaining nine species were further subdivided into three groups according to whether they were foodborne, of environmental origin or human-associated of which the latter group contained the majority of species. In addition to *C. albicans*, *C. parapsilosis* and *C. tropicalis*, which are considered human-associated species [47,48], this group also included *Malassezia restricta*, *Meyerozyma guiliiermondi* and *Wickerhamomyces onychis* (Table 1). The most persistent and re-occurring species able to grow at 37 °C were *C. albicans* (7 infants), *C. parapsilosis* (11), *D. hansenii* (5), *M. restricta* (4) and *S. cerevisiae* (5).

### 3.4. Fungal Community Dynamics

To determine how fungal diversity changed over time, the longitudinal samples were subdivided according to time of collection. The analysis showed that as the infants aged and their diet changed species diversity (as taxa/sample) increased, albeit gradually (see Appendix A). This was irrespective of whether all 29 samples were included, or if the analysis was restricted to the infants for which three longitudinal samples were available. The increase in diversity was most marked in the seven infants from whom three samples were obtained, with the mean number of taxa per sample increasing from 14.4 at 6 months, to 20.6 at 18 months with the largest range in diversity being at 12 months (range: 5 to 37 taxa/sample) (Figure 4). Due to the small infant cohort size, limited number of samples, and difference in sequencing depth between samples, the observed increase in fungal diversity did not reach significance. However, when the analyses were repeated using only samples with >10,000 reads and the data rarefied, then the alpha diversity (using Shannon metrics) still increased as the infants aged (see Appendix A).

Two notable fungi which increased in prevalence over the 12-month study period were *D. hansenii* and *S. cerevisiae*. While both species were present at all three timepoints, the majority of infant samples in which they were detected, either individually or together, were at the 12 and 18-month timepoints (*D. hansenii*, 11/14; *S. cerevisiae*, 12/15) (Figure 5 and Appendix A). The lowest abundance, of either species, was at 6 months whereas the highest abundance of either species was at 12 months. Indeed, *S. cerevisiae* was the dominant species in three of the 12-month samples, while *D. hansenii* accounted for almost all (99.4%) ITS1 reads in F1_12 (Figure 5).

## 4. Discussion

Preterm infants, with their immature immune system and an intestinal epithelial barrier of increased permeability, represent a highly vulnerable paediatric patient group. A shortened gestational period, disrupted in utero development, possible delivery by Caesarean section and significant antibiotic treatments all contribute to the bacterial composition of the preterm gut being distinct, typically characterised by reduced diversity and dominated by facultative anaerobes [22,23]. However, compared to the well-characterised bacterial community, very little is known about the mycobiome of the premature infant gut. Furthermore, while bacterial composition has been shown to differ during the first few weeks of life based on gestational maturity at birth [49], little is known about how gestational age affects and possibly shapes the enteric mycobiota of these infants.

To begin addressing this shortfall, we developed a workflow and protocols to characterise the fungal communities present in a cohort of infants, born with differing degrees of prematurity, and mapped how they develop during the first 18 months of life. Our findings reveal that the fungal communities within the preterm infant GI tract exhibit considerable diversity, were infant-specific, and characterised by the dominance of a single taxon and the presence of pathobiont *Candida* species. The makeup of the mycobiome is also influenced by GA and early life environmental factors.

Over 100 different fungi were identified with the majority classified to either the genus or species level. Despite this considerable diversity, many of the infant faecal samples contained a relatively low number of taxa, and were often dominated by a single taxon, typically a yeast from the genera *Candida*, *Debaryomyces* or *Saccharomyces*. (e.g., *C. albicans*, *C. parapsilosis*, *D. hansenii* or *S. cerevisiae*), with *Candida* colonisation influenced by the gestational age of the neonate at birth. Many of the fungi were infant-specific, often present in very low abundance, found at only one timepoint over the course of the 12-month study period and were not detected in the DNA extraction control. For those taxa for which a species identity could be determined (~70%), many were environmental fungi based on the fact they are commonly found in soil, the air, and associated with plants either as pathogens or saprophytes. Many of these fungi are incapable of growth at 37 °C making their presence in the infant samples incidental, due to either ingestion (as foodborne contaminants) or through spore inhalation and subsequent ingestion. This is perhaps best exemplified by the spore-forming species *Davidiella tassiana* (anamorph—*Cladosporium herbarum*) and *Baeospora myosura* repeatedly detected in our preterm infant cohort, although often at low abundance. Their inability to grow at human body temperature means that their presence is most likely incidental due to spore ingestion (e.g., as foodborne contaminants).

At a phylum level, the preterm infant gut mycobiome comprises exclusively of fungi from the Ascomycota and Basidiomycota phyla, with Ascomycota dominating at all three timepoints over the 12 months. This dominance was largely due to the presence of the genera *Candida*, *Davidiella*, *Debaryomyces*, *Penicillium* and *Saccharomyces*, which collectively accounted for over 50% of all fungal reads. The dominance of Ascomycota and Basidiomycota in the healthy human GI tract, of both infants and adults, has been reported previously in several studies, with these fungi also being dominant members of the skin, vaginal and oral cavity mycobiomes [10,33,50,51,52,53]. Collectively, this suggests members of these phyla are well-suited for persisting within and on the human host. In addition, the mammalian GI tract appears to be the primary ecological niche for a number of human-associated *Candida* including most notably *C. albicans*, a species rarely found in the environment [48,54].

*Candida* accounted for over a quarter of all the fungal reads and it was the most prevalent genus, being detected in every infant sample. This is in line with *Candida* species and *C. albicans* in particular being frequently identified both in the infant and adult gut [33,34,53,55,56], as well as in other body sites, including the oral cavity and vagina [50,51,57]. Six *Candida* species were identified in the preterm infant cohort, including *C. albicans*, *C. metapsilosis*, *C. parapsilosis* and *C. tropicalis*, all of which are human-associated and fungal pathobionts [47]. Indeed, at least one of these four *Candida* species was detected in every infant sample with the majority containing at least two of the species, most commonly *C. albicans* and *C. parapsilosis*. This is consistent with preterm infants having higher *Candida* colonisation rates compared to their full-term counterparts [32]. Although infrequently detected, *C. metapsilosis* was most notable for being the dominant *Candida* pathobiont in the most premature infant (25 weeks), which uniquely received the prophylactic antifungal fluconazole in the first month of life. Our study also revealed that *Candida* colonisation persists well beyond the first few weeks after birth, when many preterm infants, especially those born <28 weeks, remain hospitalised and highly susceptible to nosocomial bacterial or fungal infection. Indeed, in two of the preterm infants, *Candida* are the dominant fungi at 18 months after birth (Infant M2, *C. parapsilosis*; M4, *C. albicans*).

Although *C. albicans* is a common commensal of both the human GI tract and oral cavity [58] and is a predominant member of the vaginal mycobiome [59], *C. parapsilosis* was the most prevalent *Candida* species in the preterm infant cohort. Of the 11 preterm infants studied, eight carried *C. parapsilosis* at all timepoints over the 12 months. It was also more abundant in those infants born at <31 weeks (i.e., very/extremely preterm), whereas *C. albicans* abundance was generally higher in infants born at ≥31 weeks (i.e., moderately preterm to near term).

To date, little data is available relating to the composition of the enteric mycobiome of infants born full term. However, in a recent longitudinal study, where the anal, oral and skin mycobiomes of a cohort of term infants were characterized over the first month of life, *C. parapsilosis* along with *C. albicans* were both found to be prevalent in the anal mycobiome of these infants, present in 95% and 85% of samples, respectively [33]. This previous study also found *C. tropicalis* to be highly prevalent in these infants (present in >90% anal samples), which is in marked contrast to the results from our study, where this species was detected in just over a quarter of infant samples. A possible explanation for this marked difference could be the fact the authors found *C. tropicalis*, as well as *C. albicans*, to be present in every maternal vaginal mycobiome [33], and thus its prevalence in these infants could reflect the fact they were all born to term. This may also account for why *C. albicans*, again present in every pregnant mother, was more prevalent in the term-born infants compared to the preterm infants in our study (85% and 66%, respectively) [33].

Whilst *C. parapsilosis* is often detected in the human gut [10,33,60], it is primarily a skin commensal [61] and is often present in the hospital environment, and in particular in the neonatal intensive care unit (NICU). *C. parapsilosis* is, therefore, considered a significant human pathogen, particularly of preterm neonates. Multiple risk factors including prolonged hospitalisation, very low birth weight (<1500 g), parenteral nutrition and the use of indwelling intravenous catheters, hand carriage and indirect (horizontal) transmission contribute to the vulnerability of preterm infants to *C. parapsilosis* infection [29,62,63]. All but one of our preterm cohort were born with either very- or extremely low birth weight, and required lengthy hospitalisation after birth (mean hospital stay: 49 days). The prevalence of *C. parapsilosis* in these infants may therefore result from hospital acquisition rather than vertical transfer from mother to offspring, either before or during delivery. Considering that all preterm infants endure disrupted fetal development, with the final weeks/months of development occurring *ex utero* and with little exposure to the maternal mycobiota, this may account for why *C. albicans*, a key member of the vaginal mycobiome [59], was less abundant than *C. parapsilosis* in these infants. However, in the absence of accompanying maternal mycobiota data, we cannot discount the possibility that these findings were due to or influenced by the composition of the maternal mycobiome.

Based on the growth requirements of individual fungi, we were able to discriminate between transient versus persistent colonisers of the preterm infant gut. On this basis, only nine species possessed the ability to grow at 37 °C, representing potential stable colonisers of which *C. albicans* and *C. parapsilosis* occurred the most, and are often found in the adult human gut [33,34,53,55,56].

Three other notable persistent yeast species in the preterm infants were *D. hansenii*, *M. restricta* and *S. cerevisiae*. *D. hansenii* is a ubiquitous osmotolerant fungus, highly prevalent in dairy products and often used in the food industry as a cheese yeast [64,65]. *D. hansenii* has also been isolated from the skin and breast milk, and is a dominant member of the infant gut mycobiota during the breast-feeding period [10,65,66,67]. It has also been cultured from human faeces and some strains are reported to grow at 37 °C [64]. Collectively, this would suggest *D. hansenii* or at least certain strains of the species have the capacity to survive and colonise the preterm infant GI tract.

*M. restricta* is a lipophilic basidiomycete and, like other members of the genus, is considered to be one of the principle fungal constituents of the skin microbiota of infants and adults [52,68,69]. Thus, as a prevalent skin commensal, *M. restricta* acquisition can occur by either vertical transmission (mother to offspring) or horizontal transfer (healthcare worker to offspring). *M. restricta* has also been detected in human breast milk as well as the GI tract [10,53,60,66,67,70]. Whilst its presence in breast milk may be due to possible transfer from surrounding maternal skin, the fact it has been detected repeatedly in the human gut would indicate it can also colonise the GI tract [10,53,60,70]. This is supported by our finding that *M. restricta* is prevalent and persistent in four of the preterm infants.

*S. cerevisiae* persisted in almost half of the infant cohort, and increased in abundance over time in two of the infants, indicative of successful colonisation. The findings that *S. cerevisiae* is frequently detected in the adult human GI tract [10,48,71] and that some strains grow well at 37 °C [66,72] is consistent with *S. cerevisiae* being a stable coloniser of the human GI tract. Countering this claim for *S. cerevisiae* being a human GI tract commensal is the recent finding that *S. cerevisiae* could not be detected in stool samples of four healthy Western adults fed a *S. cerevisiae*-free diet [73]. This finding and study suggests it is a transient coloniser of the human GI tract, with its presence being dependent upon sustained consumption of food containing *S. cerevisiae* (e.g., bread) [48]. Further studies are required to establish the true nature of the relationship between *S. cerevisiae* and the human mycobiome.

Overall, the diversity of the preterm infant gut mycobiome is low, with community composition differing between infants. Similar findings have been reported for full term infants [10,33] suggesting that in early life the infant gut is populated by a relatively small number of fungal species, irrespective of whether the infant is born premature or to term. Through longitudinal sampling, we observed that as the infants aged, so the diversity and species richness of their respective gut mycobiomes also increased, albeit gradually as noted in full term infants [10]. Diet is likely to be a major factor and driver of this increased diversity. Between 6 and 9 months of age, most babies typically begin weaning (or complimentary feeding), a time when they are first introduced to solid foods. At this point in development, and over the coming months, not only does their diet increase in complexity, from purely milk-based, but these infants are also exposed to a variety of different foodborne microbes including fungi. The persistence of *D. hansenii* and *S. cerevisiae* that are both closely associated with food in our cohort highlight the importance of the post-weaning infant diet and the role it can play in fungal acquisition and in shaping the early life preterm infant gut mycobiome.

## 5. Conclusions

This study describes the development of a protocol and pipeline workflow for the characterisation of the human faecal mycobiome and its application to provide the first detailed insight into the fungal communities present within the preterm infant gut in early life. Our findings show how colonisation of the preterm infant gut is influenced by gestational age, and how age and diet impact on the subsequent development and diversity of the gut mycobiome. A characteristic feature of the preterm infant gut mycobiome is the prominence of the pathobiont *C. parapsilosis*, a prevalent hospital-associated fungal pathobiont. This study paves the way for further investigations of the developing enteric mycobiome and the importance and impact of gestational maturity at birth on community composition, as well as early antibiotic/antifungal prophylaxis in establishing both a healthy mycobiome and microbiome. Furthermore, while probiotic supplementation in preterm infants has been shown to have a positive impact, leading to a *Bifidobacterium*-dominated gut with a lower abundance of bacterial pathobionts, no such study has yet been carried out to investigate the effect of such treatment on the gut mycobiota of these infants.

## Figures and Tables

**Figure 1 jof-06-00273-f001:**
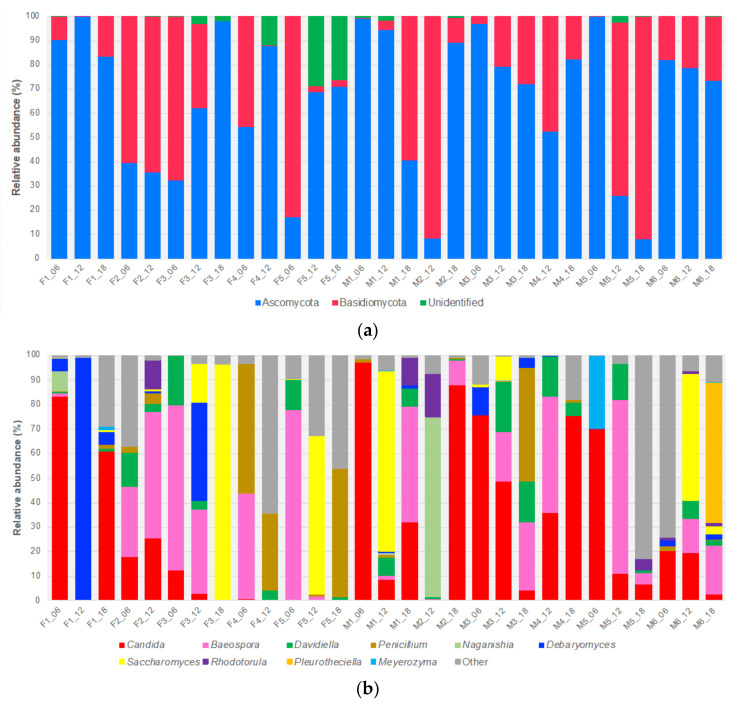
Bar charts of the relatively most abundant fungal taxa in the preterm infant gut at (**a**) phylum level and (**b**) genus level.

**Figure 2 jof-06-00273-f002:**
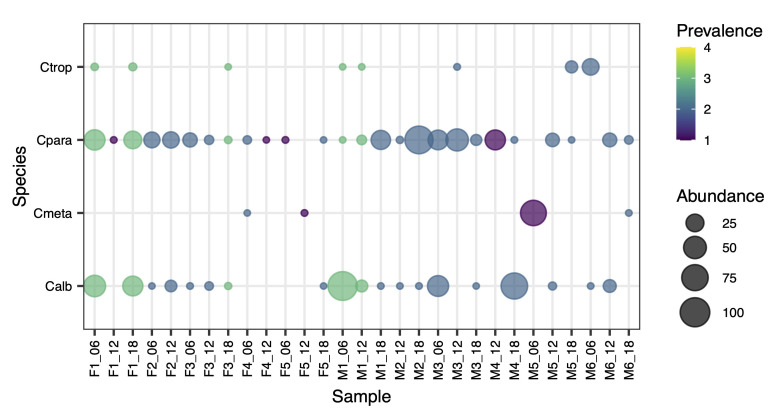
Bubble plot depicting the prevalence and relative abundance of opportunistic pathogenic *Candida* species in the preterm infant gut mycobiome (Calb, *C. albicans*; Cmeta, *C. metapsilosis*; Cpara, *C. parapsilosis*; Ctrop, *C. tropicalis*).

**Figure 3 jof-06-00273-f003:**
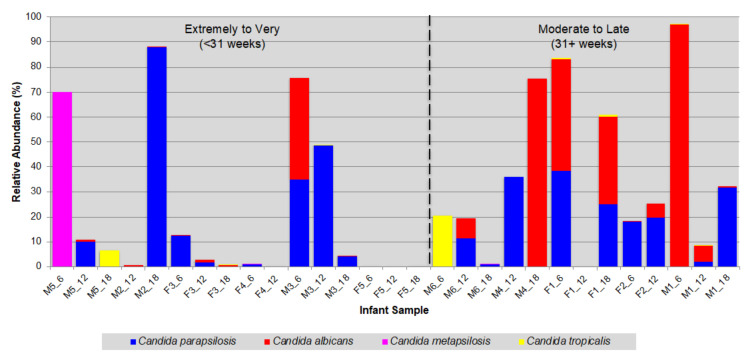
Bar graph of the prevalence and abundance of human-associated *Candida* species in the preterm infant gut mycobiome. Samples ordered according to infant gestational age at birth.

**Figure 4 jof-06-00273-f004:**
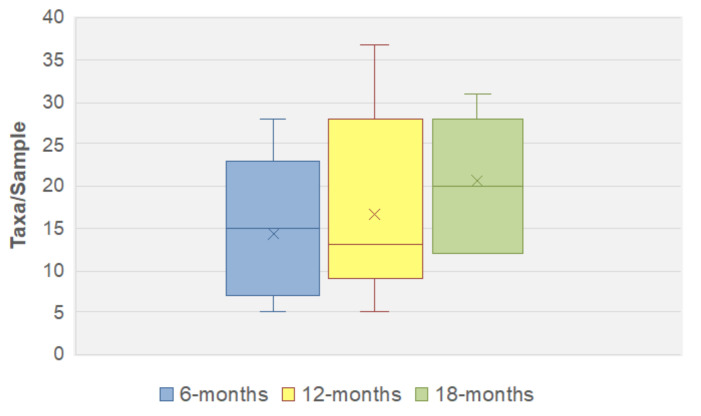
Fungal alpha-diversity depicted by a box-and-whisker plot showing number of taxa detected in each faecal sample for infants with 3 longitudinal samples. The horizontal line represents the median marker and ‘×’ represents the mean marker.

**Figure 5 jof-06-00273-f005:**
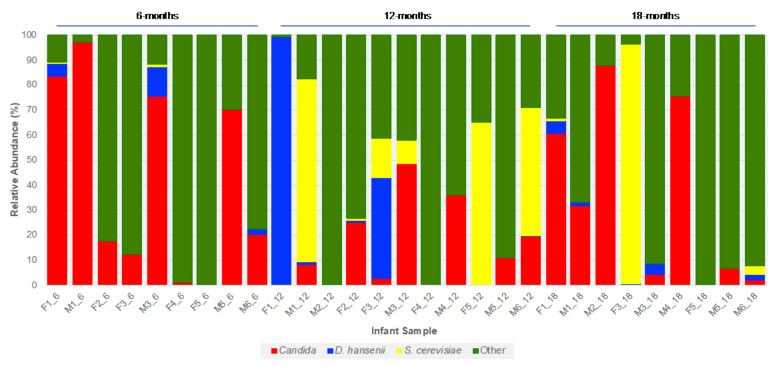
Bar chart depicting the prevalence and abundance of *D. hansenii* and *S. cerevisiae* in the preterm infant gut at between 6- and 18-months of age.

**Table 1 jof-06-00273-t001:** Persistent fungi. Species detected at two or more collection timepoints (i.e., at 6-, 12- and/or 18-months), with the ability to grow at body temperature.

Species	Category	Infant #
*Aspergillus ruber*	Environmental	1
*Candida albicans*	Human-associated	7
*Candida parapsilosis*	Human-associated	11
*Candida tropicalis*	Human-associated	2
*Debaryomyces hansenii* ^1^	Foodborne	5
*Malassezzia restricta*	Human-associated	4
*Meyerozyma guilliermondii*	Human-associated	2
*Saccharomyces cerevisiae* ^1^	Foodborne/Probiotic ^2^	5
*Wickerhamomyces onychis* ^1^	Human-associated	1

# Number of infants carrying the species; ^1^ Growth at 37 °C is strain variable; ^2^
*Saccharomyces boulardii*, a probiotic strain of *S. cerevisiae*.

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
