# Peer review of "Preterm Infants Harbour a Rapidly Changing Mycobiota That Includes Candida Pathobionts"

_jof, 2020, doi:10.3390/jof6040273_

Round 1

Reviewer 1 Report

This is a well written manuscript providing interesting data about the composition of the mycobiome in infants born preterm.  Although only 11 subjects are included, collecting samples over the first 18 months provides important longitudinal insight into how these organisms persist or change over time, and therefore provides a novel contribution in this area.  A few comments:

  1. One patient was treated with fluconazole.  Was this prophyalctic or treatment for a documented infection?
  2. The majority of infants received probiotics.  Are there data available about how probiotics may impact the mycobiome?  Any speculations to offer related to your subjects?
  3. Although data related to the mycobiota in term infants is mentioned, a somewhat more detailed comparison of the data from this study to term infants would likely be of interest to the readers.
  4. There are no data points visible in Figure 2 (at least in the version of the manuscript available to reviewers).
  5. Could the finding of environmental fungi incapable of growth at 37 degrees be attributed to contamination of the samples themselves rather than ingestion through food as suggested, or do the control samples make this possibility less likely?

Author Response

  1. One patient was treated with fluconazole. Was this prophylactic or treatment for a documented infection?

Only infant M5 received prophylactic fluconazole as he was categorised as a ‘high risk’ preterm baby based on his gestational age at birth (<27 weeks) a low birthweight of <750 g. All such infants at the Norfolk and Norwich University Hospital routinely receive prophylactic antifungal treatment via an indwelling central venous catheter. This information has been incorporated into the text and added to the infant metadata (Table S1).

  1. The majority of infants received probiotics. Are there data available about how probiotics may impact the mycobiome? Any speculations to offer related to your subjects?

Probiotic supplementation has been shown to have a positive impact on the preterm infant bacterial microbiome (e.g. Reference 35). However, to the best of our knowledge no such study has yet been undertaken to look at the effect of such treatment on the mycobiome. Furthermore, while we believe it likely that such treatment could have an impact on the early life mycobiota, without access to preterm infants who had not received probiotic supplementation, it is difficult to say with any degree of certainty what that impact might be. In a recent study by Schei et al. 2017 (Reference 10), pregnant mothers given milk containing probiotic bacteria (incl. Lactobacilli spp. and Bifidobacterium animalis) had significantly increased fungal DNA concentrations. This implies that probiotic supplementation had an impact on the infant mycobiome. Any effect of probiotics on the infant gut mycobiome would depend upon how established the microbiome is, its composition, and the origin of enteric fungi and whether it is maternally derived or acquired from the hospital environment.

  1. Although data related to the mycobiota in term infants is mentioned, a somewhat more detailed comparison of the data from this study to term infants would likely be of interest to the readers.

To date, very little data relating to the gut mycobiota of term infants exists. However, we have incorporated results from a recent 2018 study by Ward and colleagues (Reference 33) in which they characterised the early life anal, oral and skin mycobiomes of a small cohort of term-born infants over the first month of life.

  1. There are no data points visible in Figure 2 (at least in the version of the manuscript available to reviewers).

In the online journal version of our manuscript, all data points for all 4 Candida species and all 29 infant samples are clearly visible. Likewise, all data points are also visible in the original figure we submitted to the journal. As similar comments were not raised by Reviewer 2, this issue may be due to a possible computer programme compatibility problem.

  1. Could the finding of environmental fungi incapable of growth at 37 degrees be attributed to contamination of the samples themselves rather than ingestion through food as suggested, or do the control samples make this possibility less likely?

The vast majority of these fungi were not detected in the extraction control sample, and so it is unlikely therefore that their presence in the infant samples is due to kit contamination. Also, considering many of these fungi were detected in low abundance in single infant samples and at a single timepoint, would support food ingestion as their likely source. In a recent review (Reference 48) the authors concluded that many of the rare, low abundance fungi detected in human gut mycobiome studies are incidental and of environmental origin.

Reviewer 2 Report

The paper is clearly written and has an interesting and useful topic!

From a scientific point of view, it is of high interest! It is very interesting how the fungal profile varies from girls to boys and over the tree stated time points.

I would like to congrats the authors for their work! 

I have some minor comments, the study has some limitations. 

The authors showed state from the beginning of the study about the group - if any of the participant/mothers took any drug if they were all breastfeed or formula-fed, about their diet after weaning, if the breastfeeding was ongoing in tandem with solids eating

Line 145 - the link is not working (page not found)

Line 179. About Candida's presence. It is a normal fact! 

An important rule is that the figures, graphs, and tables need to stand alone, without needing to read the text. Please add more variables to Table 1, it is not clear what is with the 2 or more timepoints. Please state in the table caption. 

The European Food Safety Authority declares that Saccharomyces cerevisiae is presumably safe. In many cases, this strain can be found in the probiotic list. The author's state (Table 1) that S. cerevisiae is a foodborne pathogen

Author Response

  1. The authors should state from the beginning of the study about the group - if any of the participant/mothers took any drug if they were all breastfeed or formula-fed, about their diet after weaning, if the breastfeeding was ongoing in tandem with solids eating.

All details relating to whether each infant received breastmilk and/or formula during their stay at the Norfolk and Norwich University Hospital (NNUH) that was made available to us has now been included and added to the infant metadata table (Table S1). Likewise, we do not have access to any data relating to infant diet post weaning and cannot therefore comment on whether any of these infants were breastfed while taking solids.

  1. Line 145 - the link is not working (page not found)

Although this link worked for us via the journal’s website we have replaced it with the following shortened link: ‘https://github.com/quadram-institute-bioscience/bambi-its/’.

  1. Line 179. About Candida's presence. It is a normal fact!

Although Candida, and in particular, C. albicans, are frequently detected in the GI tract of healthy infants and adults, a recent review (Reference 48) estimated the carriage of C. albicans in the general population to be between 30% and 60%. In view of these modest estimates, we thought it important to highlight the fact that Candida were detected in 100% (29/29) of infant samples.

  1. An important rule is that the figures, graphs, and tables need to stand alone, without needing to read the text. Please add more variables to Table 1, it is not clear what is with the 2 or more timepoints. Please state in the table caption.

For clarity, we have now added the 6-, 12- and 18-months to the table caption.

  1. The European Food Safety Authority declares that Saccharomyces cerevisiae is presumably safe. In many cases, this strain can be found in the probiotic list. The author's state (Table 1) that S. cerevisiae is a foodborne pathogen.

 The reviewer is mistaken. In Table 1, we clearly say that Saccharomyces cerevisiae is ‘foodborne’; and not a ‘foodborne pathogen’. In considering the Reviewer’s comment that certain strains of S. cerevisiae (e.g. Saccharomyces boulardii) are used as probiotics, we have included this information as an additional footnote to Table 1.